# Modeling latent infection transmissions through biosocial stochastic dynamics

Bosiljka Tadić [1,2] *, Roderick Melnik [3,4]

**1** Department of Theoretical Physics, Jožef Stefan Institute, Ljubljana, Slovenia, **2** Complexity Science Hub, Vienna, Austria, **3** M2NeT Laboratory and Department of Mathematics, MS2Discovery Interdisciplinary Research Institute, Wilfrid Laurier University, Waterloo, ON, Canada, **4** BCAM - Basque Center for Applied Mathematics, Bilbao, Spain

* Bosiljka.Tadic@ijs.si

**Data Availability Statement:** All relevant data are within the paper and its Supporting Information files.

**Funding:** B.T. work is supported by the Slovenian Research Agency (research code funding number

## Abstract

The events of the recent SARS-CoV-2 epidemics have shown the importance of social factors, especially given the large number of asymptomatic cases that effectively spread the virus, which can cause a medical emergency to very susceptible individuals. Besides, the SARS-CoV-2 virus survives for several hours on different surfaces, where a new host can contract it with a delay. These passive modes of infection transmission remain an unexplored area for traditional mean-field epidemic models. Here, we design an agent-based model for simulations of infection transmission in an open system driven by the dynamics of social activity; the model takes into account the personal characteristics of individuals, as well as the survival time of the virus and its potential mutations. A growing bipartite graph embodies this biosocial process, consisting of active carriers (host) nodes that produce viral nodes during their infectious period. With its directed edges passing through viral nodes between two successive hosts, this graph contains complete information about the routes leading to each infected individual. We determine temporal fluctuations of the number of exposed and the number of infected individuals, the number of active carriers and active viruses at hourly resolution. The simulated processes underpin the latent infection transmissions, contributing significantly to the spread of the virus within a large time window. More precisely, being brought by social dynamics and exposed to the currently existing infection, an individual passes through the infectious state until eventually spontaneously recovers or otherwise is moves to a controlled hospital environment. Our results reveal complex feedback mechanisms that shape the dependence of the infection curve on the intensity of social dynamics and other sociobiological factors. In particular, the results show how the lockdown effectively reduces the spread of infection and how it increases again after the lockdown is removed. Furthermore, a reduced level of social activity but prolonged exposure of susceptible individuals have adverse effects. On the other hand, virus mutations that can gradually reduce the transmission rate by hopping to each new host along the infection path can significantly reduce the extent of the infection, but can not stop the spreading without additional social strategies. Our stochastic processes, based on graphs at the interface of biology and social dynamics, provide a new mathematical framework for simulations of various epidemic control strategies with high temporal resolution and virus traceability.

P1-0044). R.M. research was funded by the Natural Sciences and Engineering Research Council (NSERC) of Canada and Canada Research Chairs (CRC) Program and he is also acknowledging the support of the BERC 2018-2021 program and Spanish Ministry of Science, Innovation, and Universities through the Agencia Estatal de Investigacion (AEI) BCAM Severo Ochoa excellence accreditation SEV-2017-0718, and the Basque Government fund AI in BCAM EXP. 2019/00432.

**Competing interests:** Authors have no competing interests.

## Introduction

Stochastic processes of epidemic spreading in human society comprise a specific type of critical phenomena where the microscopic-scale interactions give raise to collective dynamics. Thus, mathematical modelling approaches are necessary to understand the nature of the process and control parameters that govern the transition from the micro- to global scale behaviours [1–3]. Recently, the critical phenomena in social systems have been researched using the concepts developed in physics of complex systems and networks [4]. Some prominent examples are the emotion spreading in online social networks [5–8], opinion dynamics [9], and constructive engagement for the collective knowledge creation [10]. A detailed analysis of empirical data of human activity online and related theoretical modelling [6, 11, 12] provided evidence that the prominent dynamical mechanisms enabling these collective phenomena lie in the self-organised criticality [13–15]. The appropriate agent-based modeling of these social phenomena [16] includes the individual emotional [5, 6] and cognitive properties [10] of the interacting agents.

In addition to social dynamics, the epidemic spreading processes involve some essential biological factors, such as the biology of pathogens, and certain health factors of individuals and groups. Recent COVID-19 data [17] on the outbreak with the new SARS-CoV-2 virus are an excellent example. In the absence of pharmaceutical interventions against the virus, social and sanitary measures remain of primary importance for controlling the epidemics. In this context, there are new traits to deal with, high latency times, rapid transmission, and the potential of the virus to trigger SARS (severe acute respiratory syndrome), which is a medical emergency. For a summary of COVID-19's unique properties on pathogenic, epidemiological and clinical issues, see [18, 19] and references therein. Statistically, about 20% of the infected individuals need hospitalisation, some of whom (making about 13% of all infected) have mild and moderate symtoms. However, 7% exhibit severe symptoms and need intensive care; they can further differentiate such that 4.7% have clinical stage of illnesses, ending up with a high fatality rate. Data from around the world confirm this overall picture with some regional variations [20–23]. On the other hand, about 80% of those infected show very mild clinical manifestations or remain completely asymptomatic, so that they recover spontaneously after about two weeks; this group of infected individuals often remains undetected and outside formal health management procedures. Therefore, mortality statistics can serve as an indicator of the actual number of infected individuals. The mortality rate ranges from 1.7% to 9% of registered cases, depending on the country or region. In addition to organisational health issues, it is hypothesized that such variations in clinical manifestations may be associated with the existence of different strains and potential genetic mutations of the virus [20, 24]. Theoretically, a virus that jumps on a new host can trigger its evolutionary development in the direction of better adaptation to human cells, which can make the pathogen progressively less aggressive towards the host [25–27]. Such mutations of the virus would be highly desirable to mitigate current epidemics. Currently, various possible mutations in the SARS-CoV-2 virus are being observed, giving rise to an open hot topic of discussion among researchers [28–30].

In the traditional modelling approaches based on mean-field equations, the standard SIR model [1] has been extended to take into account the above-mentioned features of the epidemic manifestation by distinguishing between four SIRU [31] or six SEIHR [32] infection stages and groups involved. Acronyms indicate the initial letter of "Suspected", "Infected", "Recovered", extended by the groups of "Exposed", "Undetected", "Hospitalised", that have own co-evolutionary dynamics. Recently, due to the characteristics of COVID-19 disease, a model has been extended to take into account eight different groups SIDARTHE [33]. These models with a large number of phenomenological parameters that are adapted to the actual

data, were able to describe the infection curve (increase in the cumulative number of infected individuals) as well assess the effects of social isolation on the flattening of the curve [34]. A new and promising line of research is opening up through the microscopic agent-based modeling of the epidemic processes [35]. These type of models are increasingly used for describing several specific issues of COVID-19 like epidemics [36–41].

Respiratory droplets and contacts are considered to be the primary routes of transmission of SARS-CoV-2 virus [42]. However, transmission via passive objects (fomits) as well as several other mechanisms (aerosol and fecal-oral transmission) are also reported as highly possible [18, 43–45]. In addition to hospital equipments, various passive objects can get contaminated, e.g., by contacts or respiratory droplets of an infected individual and the infection can be transmitted to a new host. This indirect transmission mechanism is becoming increasingly intriging given the reported long survival time of the SARS CoV-2 virus on different surfaces [44, 45]. Indirect exposure to the virus and the large number of undiagnosed cases in current COVID-19 epidemics underscore the importance of *latent infection transmission* as a new face of epidemic spreading. So far, this problem has remained outside the radar for standard modeling approaches.

In this paper, we develop an agent-based model that adequately describes these latent transmissions of the infection at microscopic scale and the emergence of global patterns. The model takes into account survival time of the virus and key personal characteristics of individuals, such as susceptibility to the virus and exposure time, which are crucial for the process. We simulate an open system, where the agents are generated over time through social activity fluctuations. As a proxy for social dynamics we use an empirical time series of fluctuations in activity collected from online social networks, precisely a segment of MySpace [46]. Previous studies show [8, 46] that these time series are reflecting compact off-line communities, and they closely represent correlations of fluctuations in community-related activities. Moreover, these time series have essential features of social dynamics, in particular persistent fluctuations with a typical daily periodicity (circadian cycle) [8, 46]. Hence, the underlying social structure is implicitly represented through temporal correlations and a variable intensity of time series. Using long time series of the social activity with an hourly resolution, which determines the time unit of the simulation step, our simulations span several weeks of real-time processes. In the time window, which lasts up to fourteen days after its first appearance in the process, an agent changes its state from "Susceptible" to "Exposed" and possibly "Infected", followed by either "Hospitalised" or "spontaneously-Recovered", after which it is removed from the process. During this period, an infected agent generates a number of viruses (contagious spots), which remain infective for other agents within the virus survival time. The most susceptible agents are likely to have severe symptoms; they are hospitalised and thus moved to a controlled environment. Among the remaining agents, most are asymptomatically infected carriers of viruses; thus, they contribute to the latent spread of infection within ongoing social dynamics for a long period before their spontaneous recovery occurs.

The process is presented as a growing directed bipartite graph composed of infected agents and viruses that they spread. This graph-based presentation allows us to identify the pathways of infection that lead to each infected individual and the number of hops the virus has carried from its origin to the current host. Note that chain interactions "host-virus-host" do not necessarily imply a particular social relationship between the implicated hosts. This mathematical framework enables us to simulate possible scenarios with the virus mutation. The development of the network framework based on the stochastic process is essential also due to multifaceted possible side effects and damages that the virus can have in a long run [19]. The simulated high-resolution process revealed the features of dynamic feedback in different scenarios that lead to altered course of the infection propagation. We determine the impact of the level of

social activity with/out lockdown and exposure of each individual and the mutation of the virus on the shape of the infection curves.

## 1 The model

An external input drives the system—a time series of social activity $s_t$, which introduces new agents in the process and sets the time $t$; the resolution is one hour and the total length is $t_{max}$. Thus the total number of agents is $S = \sum_{t=1}^{t_{max}} s_t$. The process is visualised as an evolving bipartite graph with a growing number of nodes and edges. Two types of nodes are the agent nodes representing humans (*Hnode*), and virus nodes representing contagious spots (*Vnode*). They are connected via directed edges from *Hnode → Vnode → Hnode* oriented in the direction of infection transmission. Besides its ID and the creation time, each Hnode possesses several other properties that can influence the process: the agent's state ("Uninfected", "Infected", "Hospitalised" and "spontaneously-Recovered"); the individual susceptibility to infections, h-factor, which is fixed by the creation of the node as a random number $h^i \in [0, 1]$, as well as its exposition time $T_e^i \in [1, T_e]$, where $T_e$ is the maximum exposition time in hours. Besides, we define the virus-host $g_v$ variable, which keeps track of the number of hops of the virus from its origin (the first infected node in the system) up to the current *Hnode*. A new *Vnode* can be produced at every time step by an active carrier (infected agent) proportionally to the severity of its infection, which is measured by its susceptibility; the virus node remains active for a fixed number of hours $T_v$. Over time, an agent can experience the transition from the "Uninfected" to "Infected" state, followed by one of the possible scenarios, depending on the agent's susceptibility level, $h^i$. Precisely, the highly susceptible agent (whose $h^i > 0.8$) is likely to have a severe illness, and its state will change to "Hospitalised" after a random period between two and seven days. Meanwhile, the less susceptible agents ($h^i \leq 0.8$) represent the asymptomatic cases that will stay mildly infected and unreported until eventually reaching the state "spontaneously-Recovered" after fourteen days. After changing its status to "spontaneously-Recovered", an agent is removed from the dynamics. Similarly, we remove the agent when its status has changed from "Infected" to "Hospitalised"; even though the hospitalised individuals can transfer infection, e.g., when the so-called nosocomial transmission (via health-care objects) occurs, it takes part in a controlled hospital environment. Meanwhile, the free active virus carriers who take part in the social dynamics primarily contribute to the latent infection transmission. See schematic flow in Fig 1.

The simulations start with one infected agent, placed on the active carriers list *Hactive*. At each step, an agent in the *Hactive* list can produce a new *Vnode* with a probability proportional to its susceptibility factor $h^i$. In this way, we take into account different infectiousness of symptomatic and asymptomatic individuals, which is in agreement with the average viral loads measured in the upper respiratory tract for mild and severe courses of the disease [47, 48]. Thus, more susceptible agents, who are likely to have severe symptoms, can spread viruses more often than those who are barely ill. The fluctuating time series introduces $s_t$ new uninfected agents at each time step $t$. By creating an agent, we identify its creation time $t^i$ as current time $t$ and fix its individual susceptibility factor $h^i$ and the exposition time $T_e^i$. The agent is placed on *Hexposed* list where it remains for $T_e^i$ hours exposed and can get infected by currently active viruses with the infection rate $\lambda_t$. Apart from the agent's susceptibility factor, the infection transmission rate $\lambda_t$ depends on several factors and fluctuates in time, as explained below, see Eq (1). Once infected, the agent is removed from the list of exposed agents and placed to the list of active carriers *Hactive*. We keep the identity of the virus that infected the agent and update the number of hops that the virus passed till that infection event occurs (the virus generation $g_v$). The time step ends up with revising the contents of each list. The difference

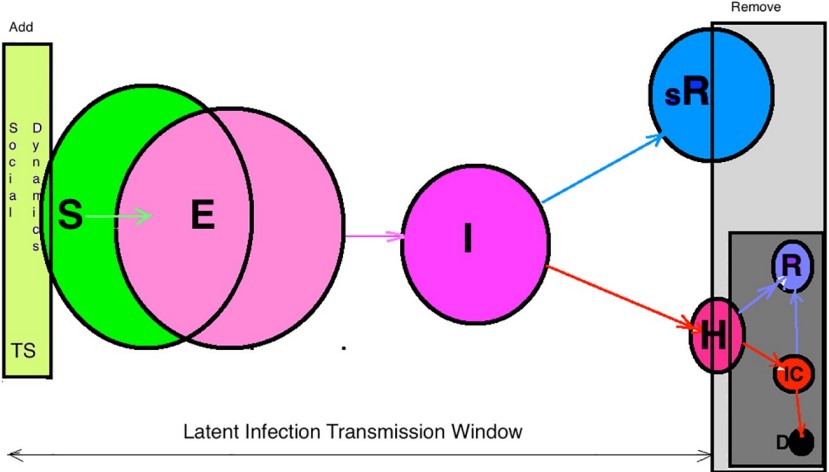

**Fig 1. Different groups of agents and transition between them.** In the simulation time window between the addition of agents through social dynamics time series and their removal by either spontaneous recovery or hospitalisation, are schematically indicated: "Susceptible" (S)→ "Exposed" (E) →"spontaneously- Recovered" (sR) or → "Hospitalised" (H). For completeness, further differentiation of the "Hospitalised" group to "Intensive-Care" (IC), "Recovered" (R) and "Deceased" (D) groups are shown in the darker square on the lower right corner; these latter subgroups do not contribute to the dynamics studied in this work.

between the current time and the node's time $t^i$ is computed for each node on a given list. Then the node is removed according to the criteria described above. The detailed program flow is described in the S1 File.

The transmission rate $\lambda_t$ varies in time and depending on the actual contact between the agent and the virus; apart from a constant (empirical value) $\lambda_0$ it is given by

$$\lambda_t^{i,v} = \lambda_0(\phi_t + 1)h^i g(g_v) \ . \tag{1}$$

The impact of the current fluctuation of the number $V_a(t)$ of active viruses, i.e., $\phi = dV_a(t)/dt/H_a(t)$ is normalised by the active number of carriers $H_a(t)$, which represents the upper limit of the new viruses at that instant of time. In addition, the infection rate is proprotional to the individual susceptibility $h^i$ of the agent in question. As stated above, our network framework allows us to follow the sequence of the virus transmission along the chain of infection events, see Fig 2, the virus generation $g_v$. Thus, we can consider the impact of hypothetical virus mutation along the chain. For this scenario, we note that by passing through a new agent node, the virus $g_v$ increases by one starting from $g_v = 1$ at the first infected individual. We assume that its sufficient weakening can be described as $g(g_v) = 2/(1 + g_v)$. Alternatively, we simulate the case without the mutation, i.e., by fixing the factor $g = 1$. The developed methodology readily allows the analysis of other scenarios as well.

## Results

In the simulations, we fix the parameters $\lambda_0 = 0.23$, $T_v = 4$ hours, the hospitalisation (2-7 days) and spontaneous recovery time (14 days) as well as the threshold susceptibility $h = 0.8$ according to the reported empirical data from SARS-CoV-2 epidemics, described in Introduction. By differentiating between the mutation ("gen") and non-mutation ("g1") scenarios, we change the maximum exposition time $T_e$ and control the intensity of the social dynamics (average number of new uninfected agents) by choosing the corresponding time series. We compute the time-evolving number of created agents and virus nodes, the edges between the infected

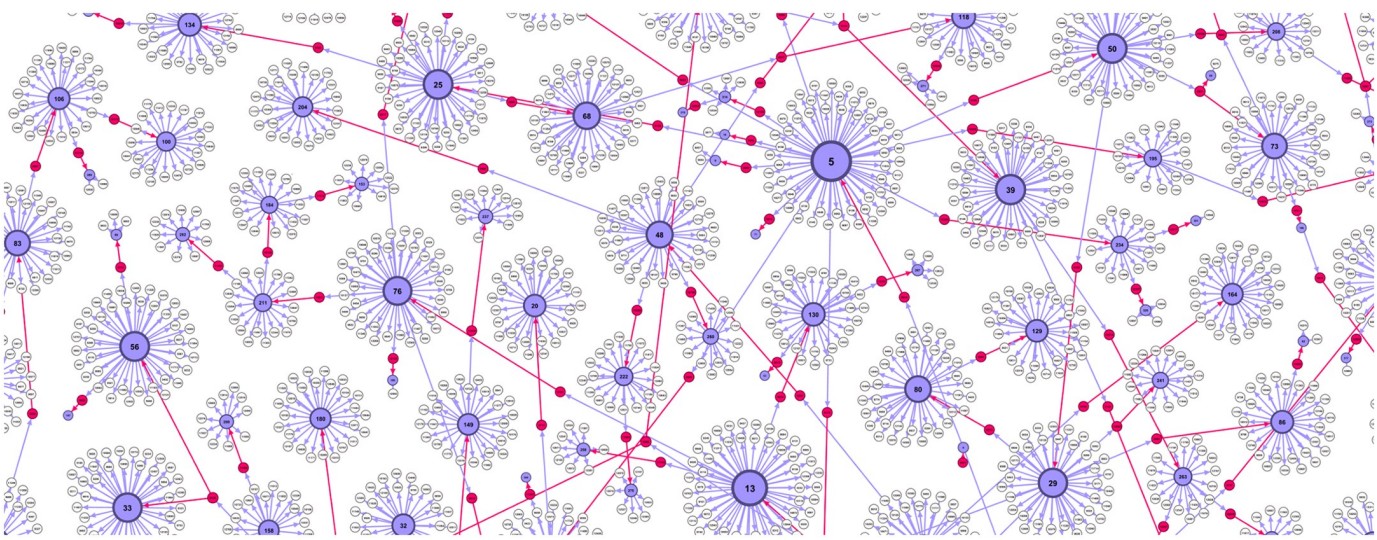

**Fig 2. Zoom-in the directed bipartite network.** *Hnodes* (blue) and created by them *Vnodes* (pale) within the first 72 hours ($\lambda_0 = 0.23$, with mutations). Infection transmission occurred through virus nodes, which are indicated by red colour, in the direction of the edge. The amount of *Vnodes* emitted by one *Hnode* is proportional to its susceptibility factor and the duration of its stay among active carriers. Among white *Vnodes* are recent ones that still can infect a new exposed agent; otherwise, each *Vnode* is non-infective four hours after its appearance.

agents and viruses, the effective transmission rate, the number of active carriers and the number of active virus spots, as well as the number of exposed and the number of infected individuals per time step. We also determine the cumulative number of infected, hospitalised and spontaneously recovered individuals for $t_{max} = 1364$ hours (56 days). Furthermore, we simulate the lockdown scenario with two types of social dynamics for the total period of 1680 hours, and a lockdown–recovery scenarios for $t_{max} = 3044$ hours. These results are presented in the following three sections.

## Infection transmission network and sampled quantities

In Fig 2, we show a part of the bipartite network that embodies the infection transmissions during the first three days. Agents (blue nodes) are enumerated by order of appearance (addition to the network), starting from the first infected agent. Along with the outgoing links from each agent node, we have *Vnodes* (pale colour) that were emitted by the agent during its active infectious period. Some of the *Vnodes*, shown in red colour, appear to infect another agent along the red edge. Meanwhile, the majority of other *Vnodes* are no longer infectious, excluding recently posted ones, which are still infectious and can change the colour to red by connecting with a new uninfected agent. Given that an agent can become infected only once during the period of interest here, the actual network has a tree structure; thus, the uninfected *Hnodes* remain disconnected from this graph (not shown).

Following the creation of new agents from a given time series $s_t$, we sample several time-dependent quantities, as shown in Fig 3 and in the following figures. We determine the time fluctuations in the number of infected and exposed agents, the number of active carriers and active viruses (see a sample data in S1 Text), as well as the cumulative infectious curve and the number of spontaneously recovered and hospitalised agents. Notably, in Fig 3, we describe the differences between the cases with and without virus mutations, meanwhile, the input time series and the other parameters ($\lambda_0 = 0.23$, $T_e = 24$ and $T_v = 4$ hours) are kept the same. In Fig 4, we show how the effective transmission rates vary in these two cases. Even though the

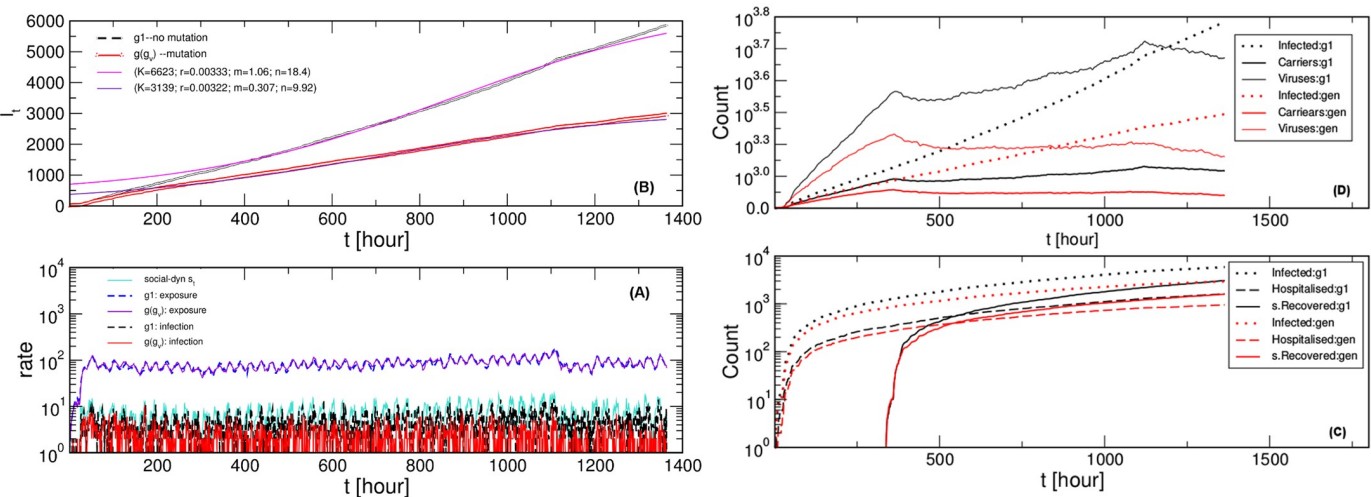

**Fig 3. Comparative simulations for virus mutations ("gen") and without mutations ("g1").** Results are for the same social dynamics–time series $s_t$ depicted in panel (A). Time fluctuations of the number of exposed agents and the number of infected agents per hour, panel (A), and the corresponding infectious curves–the cumulative number of infected agents, panel (B). Fits according to the logistic function (parameters shown in the legend). The cumulative number of spontaneously recovered and hospitalised agents are shown in panel (C) and the respective fluctuation of the number of active carriers and active viruses, in panel (D). In panels (C) and (D), the infection curves from panel (A) are shown by dotted lines, for comparison.

number of exposed individuals is practically the same, the number of infected ones per hour is smaller in the case with virus mutations than without mutations, leading to the two different cumulative infection curves in the panel (B). A similar difference then occurs in the fractions of hospitalised and spontaneously recovered cases, as shown in the panel (C). Computing the temporal variations in the number of active carriers (infected agents) and the number of active viruses nodes completes the picture. As shown in panel (D), these quantities are significantly higher in the case without mutations. It should be stressed that, given Eq (1), the fluctuations in these quantities have dynamical feedback to the effective transmission rate. We note that the obtained infection curves can be fitted by logistic function with different parameters, see more in the following section.

As the infection network in Fig 2 shows, the infection path can be followed forwards along with the directed links via red virus nodes. It appears that the number of agents that get infected from a given previously infected one varies from agent to agent, as it is shown in the bottom panel of Fig 4. Averaging over the infected agents in a given period, we get the values $<k>_{nni}$ that are larger than one, see the legend. Note that this is a quantity similar to what is in the epidemiology literature known as "R-factor" [49, 50]. Our network presentation of the process clearly shows that the R-factor is given by the relative ratio between two successive layers of the *Hnods*. Recently, studies have shown that the reproduction rate is a local measure with a limited predictive value, see the discussion and the empirical data analysed in [49–51].

## Influence of social dynamics and exposure times on the course of the infection curve

To assess the impact of the intensity of social dynamics to the infection curve, here, we simulate the scenarios with the social lockdown, which is modelled by another time series; a representative example is shown in the top panel in Fig 5. Specifically, starting with a moderately high social activity (we use the same time series as in Fig 3), the process lasts for six weeks, then the input time series is changed. Here we chose another empirical time series

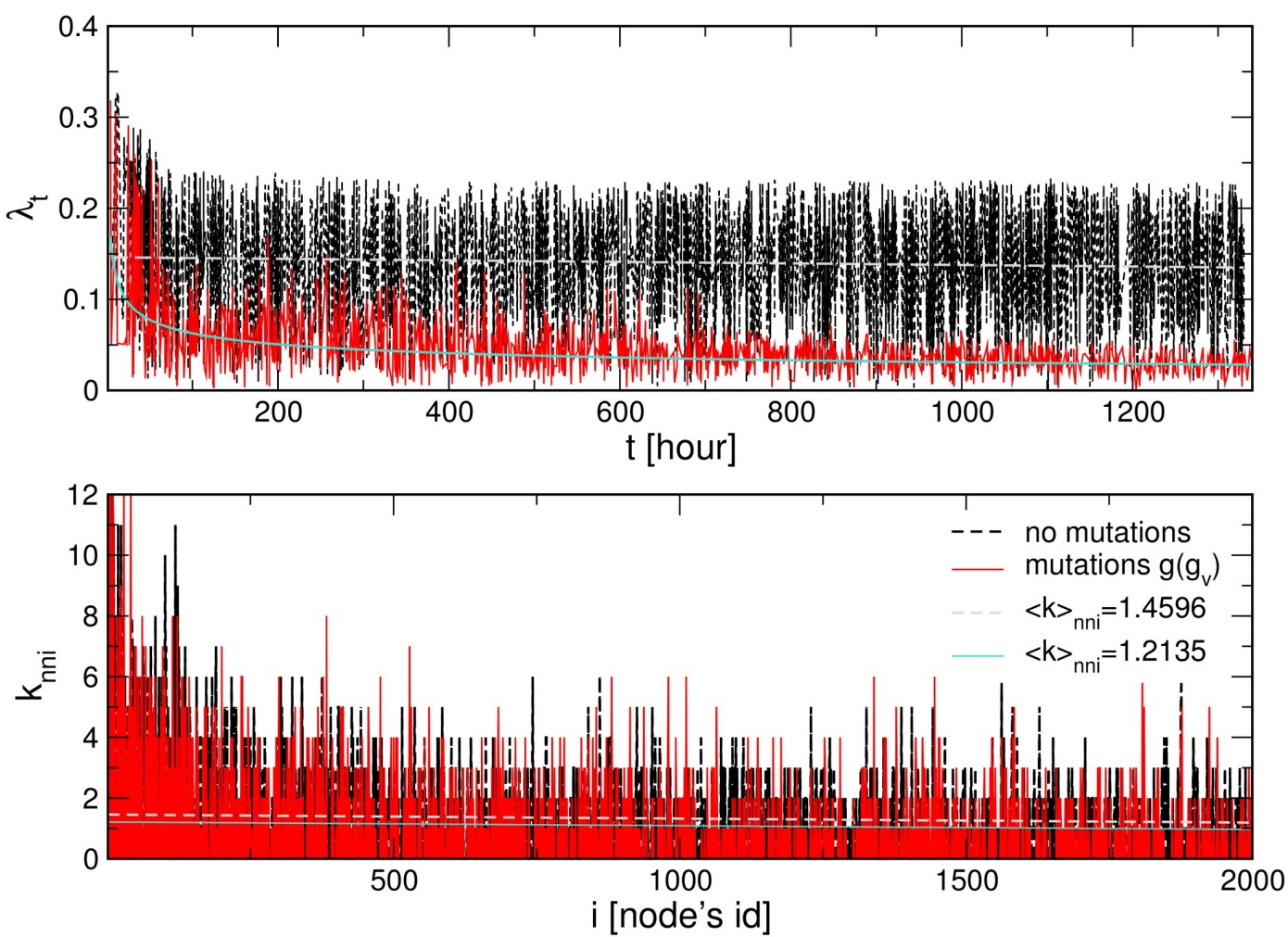

**Fig 4. Effective transmissibility.** Top: Transmission rate $\lambda_t$ vs time $t$ for the case with the virus mutations (full, red curve) and without mutations (dashed, black curve) for $\lambda_0 = 0.23$ corresponding to the course of the process in Fig 3. Bottom: Related to these transmission rates, the number of infected followers $k_{nni}$ of a given *Hnode i* indicated along the horisontal axis. For clarity, only the first 2000 nodes are shown.

(corresponding to the negative-emotion activity in MySpace data set [46]), which exhibits about four times smaller intensity but also almost absent correlations (the Hurst exponent is close to 0.5). Some simulation results are shown in the main panel of Fig 5 for the case with the mutations. These results reveal how the reduced social activity leads to an effective flattening of the infection curve, in qualitative agreement with the overall empirical data. (For easier comparison, the parameters are such that the initial part of the bottom curve corresponds to the lower curve in panel (B) of Fig 3). However, the plateau level results from the course of the entire curve from the beginning the infection. In this way, the impact of social lockdown depends on other factors that are built into the infection process before the intervention. In this figure, we further demonstrate how the increased exposure time of the agents leads to the increased level of the plateau, with other parameters fixed. In the following Fig 6, we show comparative results with/out mutations and two different lockdown scenarios, while keeping the exposure fixed.

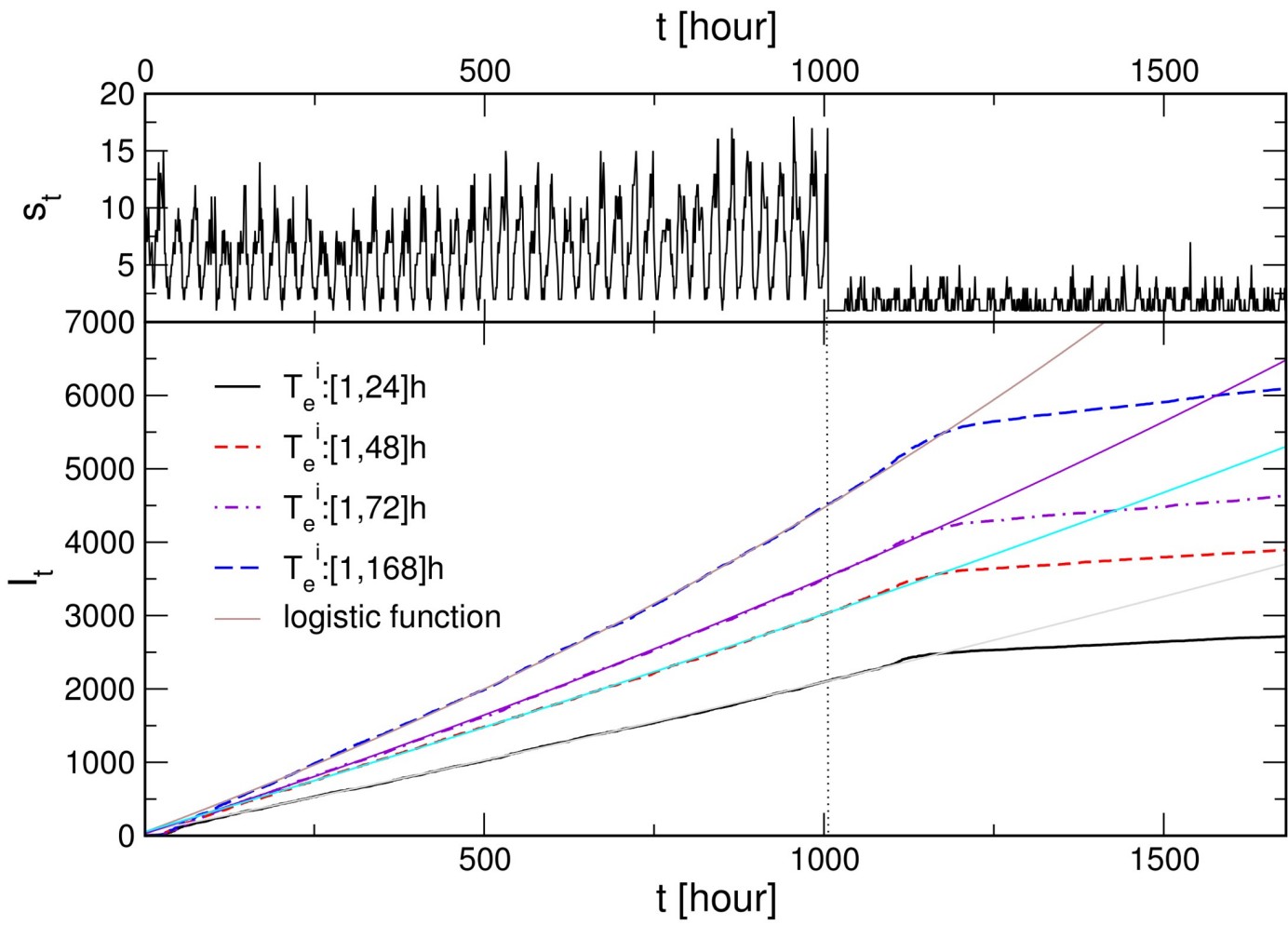

**Fig 5. Moderate social dynamics with a lockdown.** The fluctuations of social activity $s_t$ (top panel) and corresponding infection curves $I_t$ vs time $t$ for different range of exposure times, indicated in the legend (bottom), for the case with virus mutations.

To predict the extent of the infection [33, 49, 52], the course of the infection curve is standardly fitted by the sigmoid (logistic) function

$$I(t) = K\frac{1 + me^{-rt}}{1 + ne^{-rt}} \quad . \tag{2}$$

The parameter $K$ is the so-called "carrying capacity", and $r$ is the rate by which the curve reaches it, while the parameters $m$ and $n$ relate to the beginning and position of the inflexion point. The rationale behind the occurrence of this functional form (and the related derivative, the bell-like function of the infection rate curve) lies in the very nature of the infectious spreading in a *given population size*, a fraction of susceptible individuals that will be infected. In the beginning, the infection spreads to ever-larger number having practically unlimited resources. When the accelerated growth reaches the maximum infection rate (the inflexion point of the infectious curve), the process starts experiencing the limited space: the number of potential susceptible individuals that are not yet infected is reducing. Consequently, the infection rate starts decreasing while the cumulative number of infected cases asymptotically approaches the final capacity $K$.

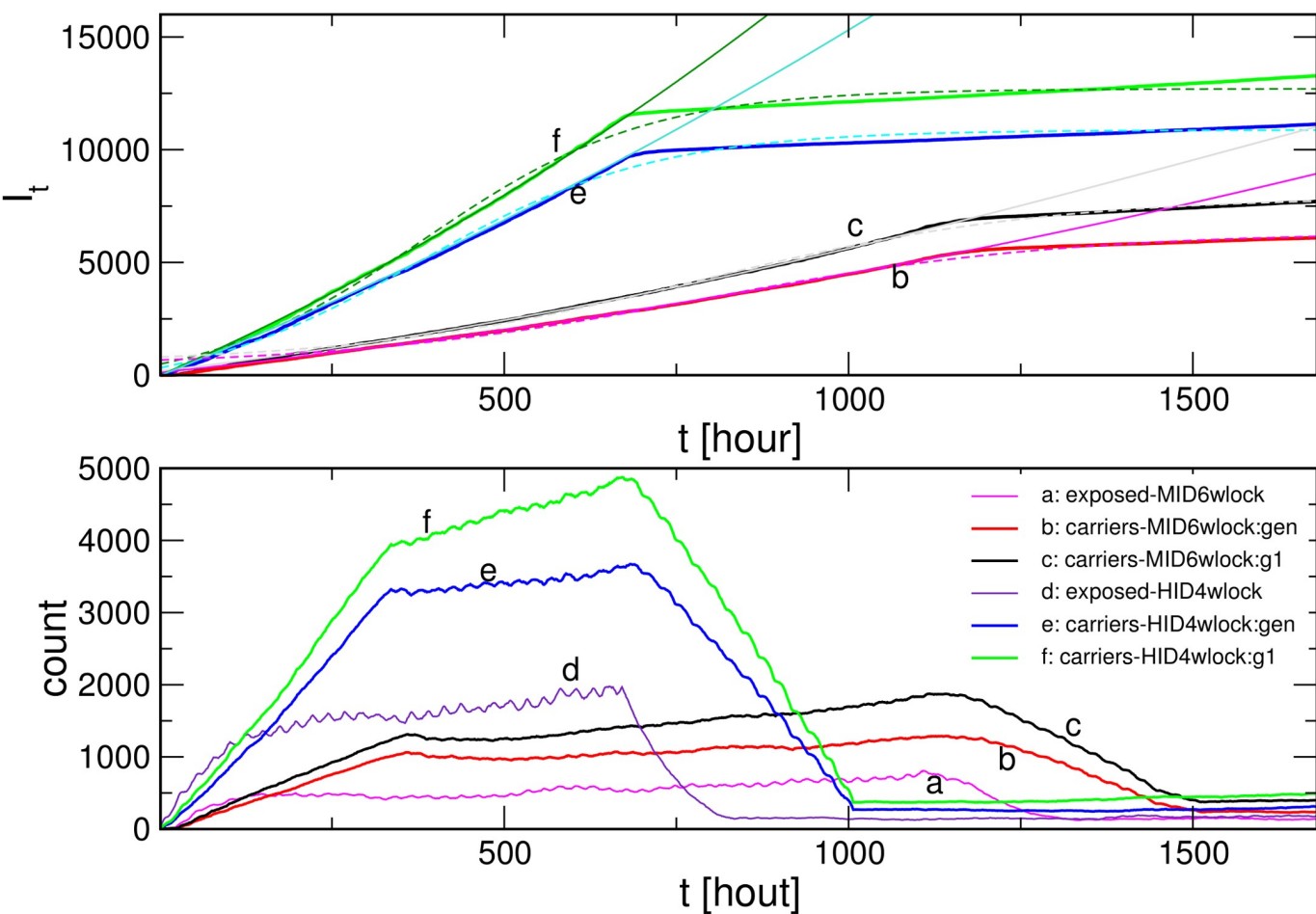

**Fig 6. Impact of the social dynamics preceding a lockdown.** Top: Infection curves for high-intensity dynamics and lockdown after week 4, and low-intensity dynamics with the lockdown after week 6; two lines in each case are the scenarios with/without virus mutations. Fits according to the logistic function (see text). Bottom: Temporal fluctuations of the number of exposed agents and the number of carriers and the number of active virus spots corresponding to the infection curves in the top panel.

Moreover, all simulations demonstrate that lowering the intensity of social dynamics, even if for a small factor such as natural day-night fluctuations, will affect the process, but with a delay. A particularly considerable lockdown comes in effect only with a ten days delay, cf. Fig 5. A closer investigation reveals, see Fig 6 that, after the lockdown, the number of exposed agents starts decreasing, reaching the corresponding lower level after the period comparable to $T_e$. Then the number of active carriers takes about a week to ten days longer to reduce and adjust to the lockdown dynamics. The number of active viruses follows this curve with a small delay (4 hours).

## Predicting the course of events after lockdown is lifted

In the literature, the impact of the imposed social lockdown during COVID-19 pandemics has been investigated from several different angles. For example, apart from economic issues, diverse social and psychological factors have been reported [53], which manifest in altering the social dynamics after a particular lockdown is lifted. In this context, our model allows us to simulate different scenarios. Notably, Fig 7 presents the results where we simulate the impact of the social activity after the 4-weeks lockdown, meanwhile keeping all other parameters at

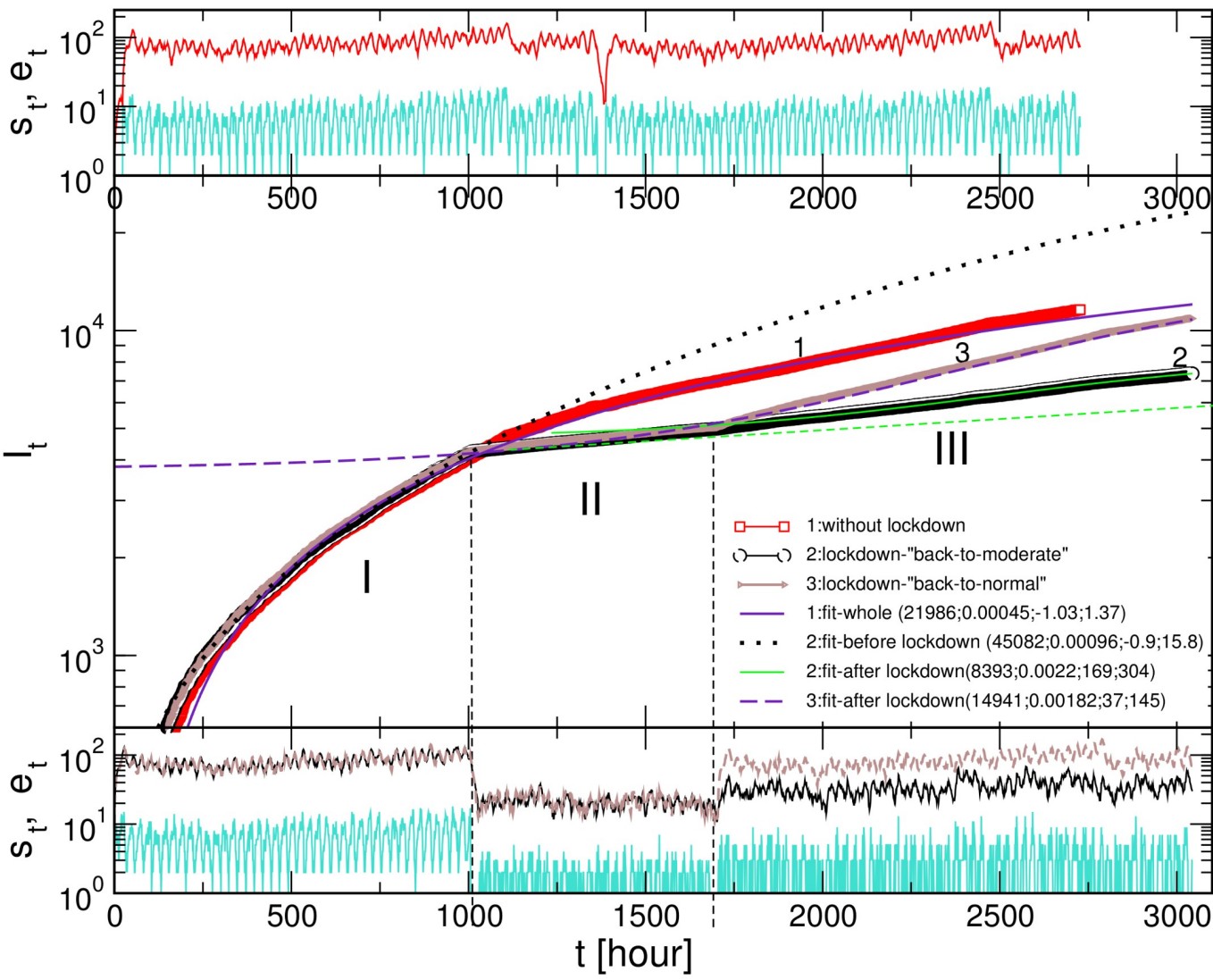

**Fig 7. Simulations of exiting scenarios.** Central panel: Infection curves $I_t$ vs time for the extended period with the naturally fluctuating social dynamics, curve (1), and for two scenarios with the lockdown intervention followed by moderate-intensity dynamics (2) and "back-to-normal" scenario (3). Periods before, during, and after lockdown are indicated as I, II, III. The fluctuating intensity of the susceptible $s_t$ and exposed $e_t$ agents corresponding to the case (1) are shown in the top panel, and to the curves (2) and (3) in the bottom panel. The last part of the signal $s_t$ in the case "back-to-normal" corresponds to the second half of the signal in the top panel.

the same level. In one case, a moderate social activity takes part, which also does not have prominent dynamical correlations (here, we amplified the time series that characterises the lockdown). In the other scenario ("back-to-normal"), we use the same time series as in the period before the lockdown and continue the process for another eight weeks. Note that this time series contains a reasonable level of dynamic correlations, as explained above. For comparison, we also simulate the situation without lockdown keeping the original level of the social activity for the period corresponding to 16 weeks of real-time. The results showing the course of the infection curve are given in the central panel of Fig 7. The driving social dynamics time series and the corresponding number of exposed agents for the scenario without lockdown are shown in the top panel and for two scenarios with the lockdown, in the bottom panel of Fig 7. These results reveal that the infection curve increases after the lockdown in all cases, but the

increase rate is low for the case of moderate social activity. In the "back-to-normal" case, the increase is much faster even if compared to the corresponding segment of the curve "1", but slower than at the beginning of infection process before the lockdown. Fitting the segment III of the curve gives the overall capacity which is still lower than the projection from the fit of the curve "1". It is also essential to notice that the curve "1" has a different course from the possible extension driven from fitting its first part (see black dotted line). We can conclude that even small fluctuations that lower the social dynamics $s_t$ (see top panel) can, through the above-described feedback mechanisms, affect the course of the infection curve, thus reducing the overall projection of the infection.

## Discussion and conclusions

We have designed an agent-based model with the high-resolution dynamics of infection process that explicitly observes the survival time of viruses and personal properties of individuals who produce them. We simulate an open system which is driven by social dynamics of the involved agents, and represented by a growing bipartite graph with the agents and viruses nodes. Considering the actual SARS-CoV-2 epidemics, the model adequately takes into account latent infection transmissions by asymptomatic virus carriers as well as indirect transmissions that can occur, for example, through contaminated surfaces during the virus survival period. These processes comprise a significant part of infection transmissions by each agent occurring in the time window which can last up to two weeks since the agent enters the system through social dynamics until it spontaneously recovers or otherwise moves to a controlled hospital environment.

The developed mathematical formalism with the infection network makes it possible to trace a path along which the virus hops over different hosts before infecting a particular individual. It thus allows us to model potential mutations of the virus along its evolution path. The results revealed the key components of this biosocial stochastic process that significantly influence the course of infection spread and the predictions of outcomes. Specifically, we have found that:

- *The intensity of social dynamics* in conjunction with the individual susceptibility of each agent is crucial for the latent infection transmission. Hence, a lockdown measure becomes effective, but with a typical delay. The simulated process with a high temporal resolution uncovers the underlying mechanisms at work. Lowering the social activity level gradually reduces the number of exposed individuals until it reaches the level of new social activity; with a delay, it causes a decrease in the number of active viruses carries and the number of active viruses that they produce. The opposite trends occur by lifting the lockdown measures. Depending on the renewed social activity, the outcome can be lower than in the case of the process without lockdown. In particular, in the "back-to-normal" situation, the final projection of the number of infected is still smaller or comparable with the original one. However, it can be reached after a much more extended period. It is interesting to point out that much smaller, natural fluctuations in social dynamics (such as day-night or workdays vs weekends) that appear periodically, as in our time series, can have profound effects on lowering the slope of the infection curves.

- *The exposure time* of each individual, is another factor that can considerably increase the course of the infection curve even with a low or moderate social activity level. Thus, modifying the exposure time of individuals or groups is an additional essential characteristic to be considered in conjunction with social dynamics measures.

- *Virus mutations scenarios* towards gradually reducing the transmission rate can slow down the growth of the infection curve. Even though these virus mutations are favourable, they have no potential to stop the spread of infection without additional social strategies.

A new mathematical framework, developed in this work, provides a robust tool for the analysis of biosocial processes in SARS-CoV-2 like epidemics. Its essential new ingredients are high-resolution dynamics, open system, and the network-assisted process presentation. Meanwhile, the traditional mean-field equations and most agent-based models in the literature have limiting factors due to the fixed size of the system. On the other hand, given the infection-network representation in our model, we do not consider direct physical distance and mobility patterns of the agents. This fact is a limiting factor for some applications of the model, for example, to describe specific geographical locations and spaces with different social groups involved (see a different study in [35, 41, 54]). Instead, the social dynamics that drive the system in our model can be varied. It represents cumulative activity participation, originating from different communities and social events. Considering the biology aspects, we have neglected possible temporal variations of the viral load of a host (besides its constant susceptibility factor), which might affect the impact that a particular host has in the process. With sufficient empirical data (current studies [47, 48, 55] are for symptomatic individuals), a modification of the model can take into account the appropriate profile of the viral load for each infected agent. Moreover, possible extensions of the model are to include a specific set of connections of each agent, a kind of ego-network, and modified transmission rates inside it. Consequently, a more heterogeneous pattern of the exposure times can emerge. However, such non-random distribution of the exposure time per agent could not significantly influence its impact to the global features of the epidemics found in this work. To reveal the nature of the underlying stochastic process, here we have simulated the infection spreading from a single source. In the meantime, the spreading from different sources or different times, which may increase the slope of the infection curve, can also be considered. Lastly, as we already mentioned, the issue of SARS-CoV-2 virus mutations is an open problem that has received increasing attention of researchers in different fields [28–30]. The infection network presentation allows us to model different ways of how particular virus strains (virus-host interaction) change over time. These aspects of stochastic biosocial processes, among others highlighted here, could represent a compelling direction for further research.

## Supporting information

**S1 File. Program flow.**
(PDF)

**S1 Text. Sample output data.**
(DAT)

## Author Contributions

**Conceptualization:** Bosiljka Tadić, Roderick Melnik.

**Formal analysis:** Bosiljka Tadić, Roderick Melnik.

**Methodology:** Bosiljka Tadić, Roderick Melnik.

**Software:** Bosiljka Tadić.

**Validation:** Bosiljka Tadić, Roderick Melnik.

**Visualization:** Bosiljka Tadić.

**Writing – original draft:** Bosiljka Tadić.

**Writing – review & editing:** Bosiljka Tadić, Roderick Melnik.

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
