## [Decision Letter · Decision Letter 0]

22 Sep 2020

PONE-D-20-23763

Modeling latent infection transmissions through biosocial stochastic dynamics

PLOS ONE

Dear Dr. Tadić,

Thank you for submitting your manuscript to PLOS ONE. After careful consideration, we feel that it has merit but does not fully meet PLOS ONE’s publication criteria as it currently stands. Therefore, we invite you to submit a revised version of the manuscript that addresses the points raised during the review process.

We look forward to receiving your revised manuscript.

Kind regards,

Ning Cai, Ph.D.

Academic Editor

PLOS ONE

Journal Requirements:

"B.T. work supported by the Slovenian Research Agency (research code funding number

P1-0044). R.M. is grateful to the NSERC and the CRC Program for their support and

he is also acknowledging the support of the BERC 2018-2021 program and Spanish

Ministry of Science, Innovation, and Universities through the Agencia Estatal de

Investigacion (AEI) BCAM Severo Ochoa excellence accreditation SEV-2017-0718, and

the Basque Government fund "AI in BCAM EXP. 2019/00432""

"No, The funders had no role in study design, data collection and analysis, decision to publish, or preparation of the manuscript."

Reviewers' comments:

Reviewer's Responses to Questions

**Comments to the Author**

1. Is the manuscript technically sound, and do the data support the conclusions?

Reviewer #1: Yes

Reviewer #2: Partly

2. Has the statistical analysis been performed appropriately and rigorously? 

Reviewer #1: Yes

Reviewer #2: Yes

3. Have the authors made all data underlying the findings in their manuscript fully available?

Reviewer #1: Yes

Reviewer #2: Yes

4. Is the manuscript presented in an intelligible fashion and written in standard English?

Reviewer #1: Yes

Reviewer #2: Yes

5. Review Comments to the Author

Reviewer #1: This work develop an agent-based model for simulations of infection transmission in an open system driven by

the dynamics of social activity; the model takes into account the personal characteristics of individuals, as well as the survival time of the virus and its potential mutations. The paper is well organized. It can be seen that the proposed model reveal complex feedback mechanisms that shape the dependence of the infection curve on the intensity

of social dynamics and other sociobiological factors.

Reviewer #2: Thank you for the opportunity to read this interesting manuscript. The manuscript is very easy to read. Authors clearly describe their aims, provide clear operationalization of all variables, and the reasoning of the procedure. In general, I consider this type of simulation to have a contribution over the mean-field theory whose results are biased by the well-mixed assumption and missing the temporal dynamics in real-world epidemic spreading.

The authors propose an agent-based model to address the latent infection transmissions by introducing temporal dynamics in simulation. The most impressive and practical thing is the authors treat the human and virus as separate elements in their model. The introduction of exposure time for the human and survival time for the virus can also reflect the latent characteristic of infection transmissions. Besides, the application of parameter gv helps capture the weakening process of the virus due to evolution.

The authors used the data collected from online social networks, MySpace, as the input data regarding human dynamical fluctuations over time. However, the disease spreading in the real-world depends on contacts (either face-to-face contacts or indirect face-to-face contacts), which may significantly distinct from online social interactions between people. Rather than data from online social media, I miss a modification of this model using data concerning direct or indirect face-to-face contacts. Or, perhaps, the authors could provide a more solid justification for their using of the online social interactions in the context of infection transmission.

As a more specific remark, I would have expected clarification (or supporting materials) of this assumption: “thus, more susceptible agents, who are likely to have severe symptoms, can spread viruses more often than those who are barely ill.” I miss a brief description of the schematic of how virus nodes infect human nodes, or which virus node infects which human node. I also miss I miss a more robust discussion on the limitations of the model.

---

## [Author Response · Author response to Decision Letter 0]

2 Oct 2020

We have found the editor's suggestions and reviewers' comments very encouraging and instructive, and we have revised the manuscript accordingly.

Specifically, we have added a description of the social activity time series, which is selected such that it closely reflects the cumulative activity in off-line social communities. We also stated that it serves as a first proxy, having no real data sets from a specific geographical location.

We have also expanded the Discussion section to highlight the absence of physical distance and mobility of the agents that limits some applications of the model, for example, to describe epidemics at specific geographical locations and social groups involved. We also comment the potential impact that changed (more realistic) distributions of the number of contacts as well as temporal variations of the viral load may have on the process that we are simulating. Several concrete extensions of the model mentioned in this context will remain for future work. 

We believe that the presented model captures the essential features of the stochastic processes underlying SARS CoV-02 epidemics, in which the latent infection transmissions have a substantial contribution. 

Prompted by your suggestion, we have removed funding information from Acknowledgments. Therefore, we would appreciate it if you could update our Funding Statement by adding the following contents:

 B.T. work supported by the Slovenian Research Agency (research code funding number P1-0044). 

R.M. research was funded by the Natural Sciences and Engineering Research Council (NSERC) of Canada and Canada Research Chairs (CRC) Program.

All Figures are uploaded as separate .tif files (with required dimension and resolution).

We have also uploaded as Supporting Information S2 Text.dat a sample of the simulation output data for a specified set of parameters (mentioned in the main text). We have also moved the program flow from the Appendix to Supporting information as S1 File.pdf.

A detailed response to the reviewers is attached. All changes are traced in the PDF file.

*

5. Review Comments to the Author

Reviewer #1: This work develop an agent-based model for simulations of infection transmission in an open system driven by the dynamics of social activity; the model takes into account the personal characteristics of individuals, as well as the survival time of the virus and its potential mutations. The paper is well organized. It can be seen that the proposed model reveal complex feedback mechanisms that shape the dependence of the infection curve on the intensity

of social dynamics and other sociobiological factors. 

Authors response:

We appreciate the Reviewer’s time and a deep understanding of our work.

Reviewer #2: Thank you for the opportunity to read this interesting manuscript. The manuscript is very easy to read. Authors clearly describe their aims, provide clear operationalization of all variables, and the reasoning of the procedure. In general, I consider this type of simulation to have a contribution over the mean-field theory whose results are biased by the well-mixed assumption and missing the temporal dynamics in real-world epidemic spreading.

Authors response: 

We thank the Reviewer for his/her time and for very encouraging statements about our work. 

The authors propose an agent-based model to address the latent infection transmissions by introducing temporal dynamics in simulation. The most impressive and practical thing is the authors treat the human and virus as separate elements in their model. The introduction of exposure time for the human and survival time for the virus can also reflect the latent characteristic of infection transmissions. Besides, the application of parameter gv helps capture the weakening process of the virus due to evolution.

The authors used the data collected from online social networks, MySpace, as the input data regarding human dynamical fluctuations over time. However, the disease spreading in the real-world depends on contacts (either face-to-face contacts or indirect face-to-face contacts), which may significantly distinct from online social interactions between people. Rather than data from online social media, I miss a modification of this model using data concerning direct or indirect face-to-face contacts. Or, perhaps, the authors could provide a more solid justification for their using of the online social interactions in the context of infection transmission.

Authors response: 

We thank the Reviewer for pointing out these issues, which, indeed, need to be better clarified, as we do in the revised text. Specifically, in the absence of real-life data sets, we use a first proxy of the social dynamics available. Among the online social networks, the MySpace data sets are such that they most closely reflect the off-line community contacts and their mutually triggered activity, as our previous studies in Refs.[8,46] show. Moreover, these data sets contain temporal correlations (persistent fluctuations that precisely match the off-line activity fluctuations) and circadian cycles, both of which define essential features of human activity patterns. We added a brief explanation in favour of this empirical time series, as compared to the alternatives (a computer-generated time series or a random number). 

We have expanded the Discussion section to once again stress the absence of geographical location and mobility patterns of the agents, also referring to different approaches in Ref.[35,41,54], where a concrete data of such kind are collected and used within the model, aiming to predict the impact of mobility that provides contacts and transmission among different groups of the population. 

Furthermore, we pointed out the limitations of our model as well as possibilities to expand the properties of the agents. For example, the model can take into account the heterogeneity in the number of contacts along which the virus could be transmitted, possibly with a changed rate, as well as temporal variations of the viral load, as some recent empirical data indicate. These issues remain to be studied in future work. 

As a more specific remark, I would have expected clarification (or supporting materials) of this assumption: ‚thus, more susceptible agents, who are likely to have severe symptoms, can spread viruses more often than those who are barely ill.‚Äù I miss a brief description of the schematic of how virus nodes infect human nodes, or which virus node infects which human node. I also miss I miss a more robust discussion on the limitations of the model.

Authors response: 

We thank the Reviewer for pointing out this issue. Different infectiousness of the agents, especially between asymptomatic and symptomatic individuals, needs to be clarified. Recent studies, for example, in Refs. [47,48,55], found that the viral load varies during the infectious time and that the average viral load in the upper respiratory tract of mild and severe symptomatic individuals is proportionally different. Also, a summary in Ref.[18] points out that “80,9% were considered asymptomatic or mild pneumonia but released large amount of viruses at the early phase of infection”. We do not have enough data to adequately introduce another variable for the temporal variations of the viral load of each agent; therefore, we use the susceptibility h^i that effectively differentiates asymptomatic from symptomatic ones. In the Model description, we added a comment regarding different viral loads that motivates the corresponding model rule. Furthermore, we mentioned the occurrence of the temporal patterns of viral loads in the Discussion by pointing out the limitations and potential extension of our model.

In the graph, we have precisely the link along which one host infects the other one via its virus. Perhaps, the Reviewer is pointing to potential (social) relationships between the hosts, which we do not take into account in this version of the model, as above described. Thus, we randomly select among currently existing active viruses; each of them is posted by a known active carrier. In the expanded Discussion, we pointed out several limitations of the current version of the model, as stated above.

%%%%%%%%%%%%%%%%%%%%%%%%%%%%%%%%%%%%%%%%%%%%%%%%%%%%%%%%%%%%

---

## [Decision Letter · Decision Letter 1]

12 Oct 2020

Modeling latent infection transmissions through biosocial stochastic dynamics

PONE-D-20-23763R1

Dear Dr. Tadić,

We’re pleased to inform you that your manuscript has been judged scientifically suitable for publication and will be formally accepted for publication once it meets all outstanding technical requirements.

Kind regards,

Ning Cai, Ph.D.

Academic Editor

PLOS ONE

Additional Editor Comments (optional):

Reviewers' comments:

Reviewer's Responses to Questions

**Comments to the Author**

1. If the authors have adequately addressed your comments raised in a previous round of review and you feel that this manuscript is now acceptable for publication, you may indicate that here to bypass the “Comments to the Author” section, enter your conflict of interest statement in the “Confidential to Editor” section, and submit your "Accept" recommendation.

Reviewer #1: All comments have been addressed

Reviewer #2: All comments have been addressed

2. Is the manuscript technically sound, and do the data support the conclusions?

Reviewer #1: Yes

Reviewer #2: Yes

3. Has the statistical analysis been performed appropriately and rigorously? 

Reviewer #1: Yes

Reviewer #2: Yes

4. Have the authors made all data underlying the findings in their manuscript fully available?

Reviewer #1: Yes

Reviewer #2: Yes

5. Is the manuscript presented in an intelligible fashion and written in standard English?

Reviewer #1: Yes

Reviewer #2: Yes

6. Review Comments to the Author

Reviewer #1: No comments for the author, including concerns about dual publication, research ethics, or publication ethics.

Reviewer #2: I am satisfied with the way the authors have incorporated my comments. Especially, they did great work in explaining the problem of using online social media data instead of real face-to-face contact data. They also described the limitations of their study.

---

## [Editor Report · Acceptance letter]

15 Oct 2020

PONE-D-20-23763R1 

Modeling latent infection transmissions through biosocial stochastic dynamics  

Dear Dr. Tadić:

I'm pleased to inform you that your manuscript has been deemed suitable for publication in PLOS ONE. Congratulations! Your manuscript is now with our production department. 

Kind regards, 

on behalf of

Dr. Ning Cai 

Academic Editor

PLOS ONE